# A Sustainability Perspective for Unbound Reclaimed Asphalt Pavement (RAP) as a Pavement Base Material

**Christina Plati * and Brad Cliatt**

Department of Transportation Planning and Engineering, National Technical University of Athens, 15780 Athens, Greece; bcliatt@mail.ntua.gr
**\*** Correspondence: cplati@central.ntua.gr; Tel.: +30-210-772-1363

**Abstract:** The present study aims to investigate reclaimed asphalt pavement (RAP) materials for utilization for a pavement base layer material with the goal towards increasing the reutilization of materials and the movement towards increased pavement sustainability. Reduced cost for materials and transportation of materials, overall environmental benefits and many other advantages have led to increased interests in utilizing RAP in pavements including as base materials for highway/roadway construction projects. The potential advantages of utilizing RAP as an unbound base material are known; however, its overall application is still limited partially due to the lack of systematic evaluation studies for the parameterization of RAPs mechanical behavior in pavement design. With this in mind, the current investigation focuses on the resilient modulus ($M_r$) properties of RAP aggregates in terms of a material's elastic response. Experimental data from tri-axial stress tests on specimens consisting of RAP, aggregates and a mixture of both materials are investigated. A number of constitutive models for the description of mechanical behavior of RAP materials are investigated. The required procedures for determining the constitutive constants of the constitutive models is outlined for the aforementioned materials. A comparative analysis is applied, and the related results are evaluated. The main conclusion is that RAP materials can be utilized as a base material in the framework of pavement sustainability, as its behavior under loading conditions are similar to virgin aggregate (VA) materials and can be simulated by using appropriate constitutive models for pavement design processes.

**Keywords:** sustainability; pavement; RAP; base; aggregates; unbound material

## 1. Introduction

In the context of adopting more environmentally friendly techniques and towards the ultimate goal of increased sustainability, the roadway construction industry has turned its attention to the reutilization and recycling of asphaltic materials for projects undergoing pavement rehabilitation or reconstruction. This policy, in addition to the obvious environmental benefits can also contribute to the reduction of construction and maintenance costs associated with road pavements. In other words, through pavement recycling and the utilization of waste construction materials, many potential sustainable materials can be utilized as a sustainable aggregate for flexible pavement construction projects. Reclaimed asphalt pavement (RAP) is one such of these materials. The aggregate material is derived through the milling of the asphalt bound layers during pavement rehabilitation procedures.

In general, based on previous experience, the usage of RAP has been most commonly utilized for the asphalt bound layers of pavement structures; however, experience has shown that it has potential for usage as an unbound granular base and/or subbase, a stabilized base aggregate and embankment or fill material [1]. Currently, economic benefits are a main rationale for utilizing RAP material in pavement structures and, in general, asphaltic material is usually the most expensive road material. However, in the manufacturing processes of RAP mixtures, intermediate steps are required

to process the material (crushing, sieving). There are other associated costs that could influence the economics, including, for example, transportation and hauling that may also need to be considered. These additional procedures may incur additional costs that exceed the initial cost benefits. On the other hand, the cost of reduced hauling if material is re-utilized on site or economic incentives provided by many countries for the reuse of this recovered asphalt material often balance these additional costs.

In addition to potential cost benefits, the environmental benefits of reusing these materials are obvious as considerable amounts of "virgin" material are saved from usage and the environmental cost associated with the extraction and preparation of virgin material for road infrastructure projects. After the recycling of these asphalt materials, they essentially cease to be industrial waste and become a valued resource; in summation, this reuse saves considerable energy and landfill space that would be needed to handle it as a waste product. Obviously, with the use of RAP, a percentage of these asphalt materials are reused when implemented in the bound layers. Beyond this, there are energy consumption benefits from the reduced preparation and hauling involved when reutilized. Furthermore, asphalt recycling and reuse leads to limiting gas emissions (carbon footprint) that is vitally important. By utilizing certain low temperature techniques, it is possible to reduce emissions (in relation to higher temperature techniques) by up to 30% to 50%. In addition, fuel savings are increased, and gas emissions are further reduced, as excavations and transport of virgin aggregates are significantly reduced during construction processes.

The economic, environmental and design benefits for RAP usage in bound layers are known and supported by extensive research and implementation. However, the same information is currently limited in terms of RAP inclusion into unbound pavement layers. This is getting to be an increasingly important issue as the availability of virgin aggregates for all pavement layers is becoming progressively difficult to source. Thus, the necessity and desire for use of available sustainable resources is quickly increasing. Consequently, RAP materials are now being examined in greater detail for their potential utilization also in the base layers of flexible road pavements. It makes sense to expect similar economic, environmental and sustainability benefits for the utilization of RAP materials at the base layers of road pavement structures, while investigation is required into the impact on the overall pavement design procedures.

Beyond the economic and environmental benefits of RAP materials integrated into pavement structures, knowledge of their structural behavior and performance is of great importance. In regard to RAP inclusion into bound pavement layers, laboratory and in-situ investigations have shown that with the appropriate design, mixtures including RAP material can achieve bound pavement layers with similar expected lifetime and quality when compared to the same layers containing virgin aggregate materials [1,2]. Nevertheless, due to the nature of the RAP material and its potential use into unbound pavement layers, more in-depth knowledge concerning its physical and mechanical properties is required. The way these properties affect its elastic responses under loading is required information for pavement design procedures.

In general, the unbound material's elastic response under loading conditions is one of the most important parameters to identify the material. The Resilient Modulus ($M_r$) is employed to define these material responses for unbound materials in pavement structures for design purposes. The resilient modulus is significantly more complex than the static modulus. It is directly affected by the induced stress in the pavement structures, as well by environmental factors. Specifically, in order to investigate the elastic response of an unbound material, the resilient modulus of the material is most often determined through standardized testing protocols with specialized equipment. Alternatively, prediction of the resilient modulus for combinations of stresses induced into the material is possible through the implementation of suitable constitutive models. Multiple models for the prediction of unbound pavement material responses have been developed over the years. It is necessary to investigate the applicability of these multiple models on aggregates containing RAP, due to the potential variation in the properties of these materials in relation to more standardized aggregate base materials in pavement structures.

With the above mentioned in mind, the present study aims at investigating the parameterization of RAPs mechanical behavior in pavement design, when it is considered in the unbound pavement layers. More specifically, varying nonlinear constitutive models will be investigated as to their appropriateness to model the behavior of material containing RAP within the base layers of flexible pavement structures. Based on a regression analysis of laboratory determined resilient moduli, the parameters of the considered models will be defined. The investigation will focus on modeling RAP behavior in the pavement design processes in order to provide evidence in support of the material's sustainability in the construction of unbound pavement layers.

The investigation will start in the laboratory through characterizing physical and mechanical properties of the tested materials. More specifically, gradation, proctor density, optimum water content and tri-axial resilient modulus testing will be undertaken on specimens prepared from RAP, aggregates and a mixture of both these materials. Based on the outcomes of the tri-axial resilient modulus testing, the data will be statistically analyzed in order to determine the regression constants required for the investigated nonlinear constitutive models that have the potential to determine the stress/strain relationship of the tested materials. The predicted moduli will be evaluated and compared with reference to the respected measured moduli of a virgin aggregate (VA), in order to examine which constitutive model(s) are the most appropriate for predicting the RAP resilient modulus response within a base layer. The output will help model RAP behavior when incorporated as an unbound base material in pavement design methods with the goal to take advantage of the sustainable benefits that are derived from the re-utilization of this material in lieu of the less readily available virgin aggregates.

## 2. Use of RAP for Unbound Pavement Layers

RAP is the name given to asphalt concrete (AC) pavement materials that are mechanically milled and removed from flexible road pavement structures that are undergoing either rehabilitation or reconstruction. The extracted material is comprised of mainly aggregate materials (~90% to 95%) with a limited amount of bitumen (~5% to 10%) to bind the aggregate together. Recycling of this material began as early as the 1970s, with in-situ recycling of these materials into the asphalt bound layers beginning soon afterwards. During the 1980s, in plant recycling began to gain ground for the addition of materials into asphaltic materials to be utilized in the upper bound layers of pavement structures. Knowledge concerning the implementation of RAP material into the upper bound layer is now readily available, while many road agencies around the world have standardized testing and protocols for the inclusion of the material into the upper asphalt bound layer. It is worthwhile mentioning that for the bound layers there are multiple recycling techniques including: Cold in plant recycling, hot in plant recycling, cold in-situ recycling and hot in-situ recycling.

Knowledge concerning the inclusion of RAP, however, into the lower unbound layers has received less attention and less in-depth research is available. According to the currently available research, RAP into the unbound base layers can be added, either in combination with a stabilizing agent or in an un-stabilized and unbound format. Stabilization of the unbound layers with a stabilizing agent has over the years received, perhaps, the most attention. Multiple stabilization techniques are available including stabilization with the addition of cement, fly ash, foamed asphalt etc. [3–7]. Thakur and Han [8] for example studied RAP for usage as a base layer aggregate in highway construction suggesting it to be a sustainable solution. The mentioned research detailed experimental investigations on RAP bases that were treated, including RAP blended aggregates, fly ash and cement and stabilized RAP, and geocell contained RAP aggregates.

As previously mentioned, RAP inclusion into bound layers and the stabilization of RAP into unbound layers have been areas of more extensive research; however, less in-depth research has been performed on RAP inclusion into unbound base layer materials and the response expected from the RAP material inclusion. Alam et al. [9] stated that literature indicates that RAP has a structural value for usage as a pavement layer. The research indicated that limited research currently exists that quantifies its structural properties with fundamental engineering properties and that this is

more pronounced for high RAP content mixtures. The $M_r$ of unbound material is required in the mechanistic–empirical pavement design guide. Montepara et al. [10] stated that the usage of the RAP as a subbase material, even through blending together with virgin aggregates, is the focus of increasing worldwide interest due to the potential for high amounts of RAP to be recycled in comparison to other recycling techniques.

Kim et al. [11] looked into the mechanical properties of the materials and conducted $M_r$ tests for specimens with varying percentages of RAP and aggregates. The study investigated the effect on material stiffness, concluding that blended 50% aggregate/50% RAP specimens had stiffness equal to 100% aggregate specimens at lower confining pressures, while at higher confinement levels the RAP specimens were stiffer. The study provided a base for further investigation into RAP inclusion into the base layers of pavement structures. In a study by Song and Ooi [12], laboratory testing was implemented to evaluate fundamental properties including the $M_r$ concluding that the resilient modulus of 100% RAP is greater than for a virgin aggregate. Dong and Huang [13] also conducted laboratory testing to assess the $M_r$ of unbound RAP, crushed limestone, and crushed gravel prepared with a similar gradation and compaction level. Results from the study showed that RAP exhibited a higher resilient modulus when compared against the unbound aggregates. The permanent deformation, however, of the RAP material was in general higher than that of the crushed aggregates. More recent research by Cliatt et al. [14] also concluded that investigated RAP aggregate materials had $M_r$ values that are equal to or are in excess of virgin aggregates.

In regard to other investigations, in Edil et al. [15] the objective of the investigation was to describe crushed recycled concrete (RCA) and RAP properties as an unbound base without stabilization, to evaluate how both RAP and RCA behave in-situ, and to assess and evaluate the design of pavement utilizing RAP and RCA. The investigation undertaken on RAP in particular indicated that RAP materials are suitable, in general, as a material for unbound base course layers and that they have near equivalent or increased performance characteristics in comparison to virgin aggregates in regards to toughness, stiffness, freeze-thaw properties, and to wet-dry durability. Nokkaew [16] concluded that the utilization of RAP aggregates for usage in a base course showed lower distress when compared to the conventional aggregate of limestone that was utilized as a control material. This implied that the RAP aggregate was of a high quality that can be utilized in base course layers for road construction. Hoppe et al. [17] looked into the feasibility of utilizing RAP for road base applications, his review indicated a current direction to incorporate RAP to levels reaching 50% in unbound base layer blends; however, there is a lack of uniformity and limited specifications. Recent research by Ullah et al. [18] investigated the impact of alterations in gradation in regards to the permanent deformation effects by looking into a variety of prepared RAP's that was combined together with virgin aggregate for the base course. The research aimed to improve gradation curves and to establish limits for mixtures containing RAP that may result in similar or better performance than the 100% VA that is used to construct base courses.

Attia and Abdelrahman [19] concluded that the usage of RAP for a base layer material is a sustainable rehabilitation technique and it reduces cost. Appropriate characterization of the stress dependent behavior within the layer of pavement structure may significantly impact the overall accuracy of the predictions of pavement responses. Noureldin and Abdelrahman [20] investigated the use of RAP within the pavement base layer and that it should consider the impact of a multitude of factors which affect the $M_r$ in-situ. Specifically, the research looked into the appropriateness of the usage of various nonlinear constitutive models currently utilized for base layer materials. It concluded that only a limited number of models could be utilized to describe the RAP behavior effectively. The investigation evaluated multiple RAP percentages (50%, 75%, and 100% by weight).

Puppala et al. [21] investigated the sustainable reutilization of both limestone quarry fine material and RAP within flexible pavement base layers, and, in the investigation, stated that although the usage of both recycled materials and other byproducts as pavement base layer materials has gained increasing acceptance, that a comprehensive geotechnical characterization is still lacking for the

materials. Edil [22] concluded that, in comparison to conventional base layer materials, RAP has an increased modulus. In addition, he stated that it is important that sustainable construction techniques should be endorsed and that the benefits of recycled material include a reduction in energy, greenhouse gas emissions, natural resources and expenses.

Available research provides an indication that RAP material inclusion into the base layer is both feasible and sustainable; however, more research is still required to support these initial indications. In regard to specifically to the modeling of the resilient behavior of the RAP materials, very limited information is currently available concerning the evaluation and potential choice of an appropriate constitutive model(s). These areas are the focus of the current research.

## 3. Constitutive Modeling of Unbound Materials

A multitude of researchers have produced, over the years, constitutive models attempting to define the known nonlinear behavior of unbound granular materials for use in pavement unbound layers. As inputs, the models use either laboratory results on the unbound granular aggregate material resilient response under loading or the actual physical characteristics of the aggregate (gradation, shape etc.). Unbound granular materials, including RAP, are affected by many parallel factors and it has proven difficult to numerically model their behavior as the material is directly affected by a wide range of factors, including for example: Moisture content, density, gradation, fines content, aggregate type and stress states. Most of the widely accepted models today are based on results from regression analysis of laboratory $M_r$ results.

The K-θ model by Hicks and Monismith [23] is one of the most commonly implemented constitutive models due to its inherent simplicity and is still widely utilized. It was one of the first constitutive models implemented for pavement analysis when it was included in the 1986 American Association of State Highway and Transportation Officials (AASHTO) pavement design guide. Later, the Boyce model [24] expanded upon the K-θ model, took into account both the mean stress and deviatoric stress, and helped provide a basis for many future models. The Uzan model [25], in partial response to the K-θ model not fitting well with the datasets under circumstances with significant shear stresses, developed a three-parameter model that could adjust for these shear stress levels. The Pezo model [26] later took into account the effects produced on the resilient modulus incorporating the confining pressure and the imposed axial deviator stress. Still later the original Uzan model was later modified by substituting the deviator stress with the octahedral stress, with the model widely recognized as the Universal model [27]. More recently, during the update of the Mechanistic Empirical Pavement Design Guide (MEPDG) the universal model was slightly modified and is now one of the most widely utilized models due to its inclusion in the MEPDG. For the present study, the following four constitutive models are investigated to access their ability to model the resilient behavior of materials containing RAP.

$$\text{K} - \theta \text{ model } Mr = k_1 P_a \left( \frac{\theta}{P_a} \right)^{k_2}, \tag{1}$$

$$\text{Uzan model } Mr = k_1 (\theta)^{k_2} (\sigma_d)^{k_3}, \tag{2}$$

$$\text{Pezo model } Mr = k_1 (\sigma_d)^{k_2} (\sigma_3)^{k_3}, \tag{3}$$

$$\text{MEPDG model } Mr = k_1 P_a \left( \frac{\theta}{P_a} \right)^{k_2} \left( \frac{t_{oct}}{P_a} + 1 \right)^{k_3} \tag{4}$$

where: $M_r = \sigma_d / \varepsilon_r$, $\theta$ is the first variant of the tensor stress = bulk stress = $\sigma_1 + \sigma_2 + \sigma_3 = \sigma_1 + 2\sigma_3$, $\sigma_d$ is the deviator stress = $\sigma_d = \sigma_1 - \sigma_3$, $\varepsilon r$ is the recoverable strain, $\tau_{oct}$ is the octahedral shear stress, $k_1$, $k_2$, $k_3$ are regression analysis constants from laboratory determined testing and Pa = atmospheric pressure (kPa) (Figure 1).

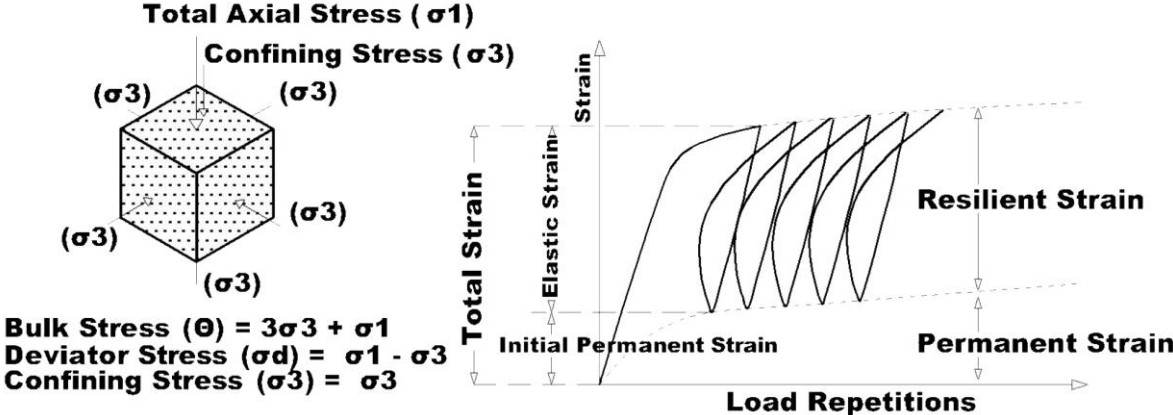

**Figure 1.** Resilient Modulus ($M_r$) and stresses.

Based on laboratory results, the aforementioned four constitutive models will be utilized to define the behavior of the materials investigated in the current research.

## 4. Laboratory Study

### 4.1. Material Description

For the investigation three materials were investigated (1) a 100% RAP material extracted from a road section undergoing rehabilitation. The material was partially graded after being extracted from the road in order to meet gradation specifications (2) a 50% RAP/50% VA mixture and (3) and a 100% VA material suitable for base layer construction and to be implemented for the same rehabilitation project. The investigated materials are shown in Figure 2.

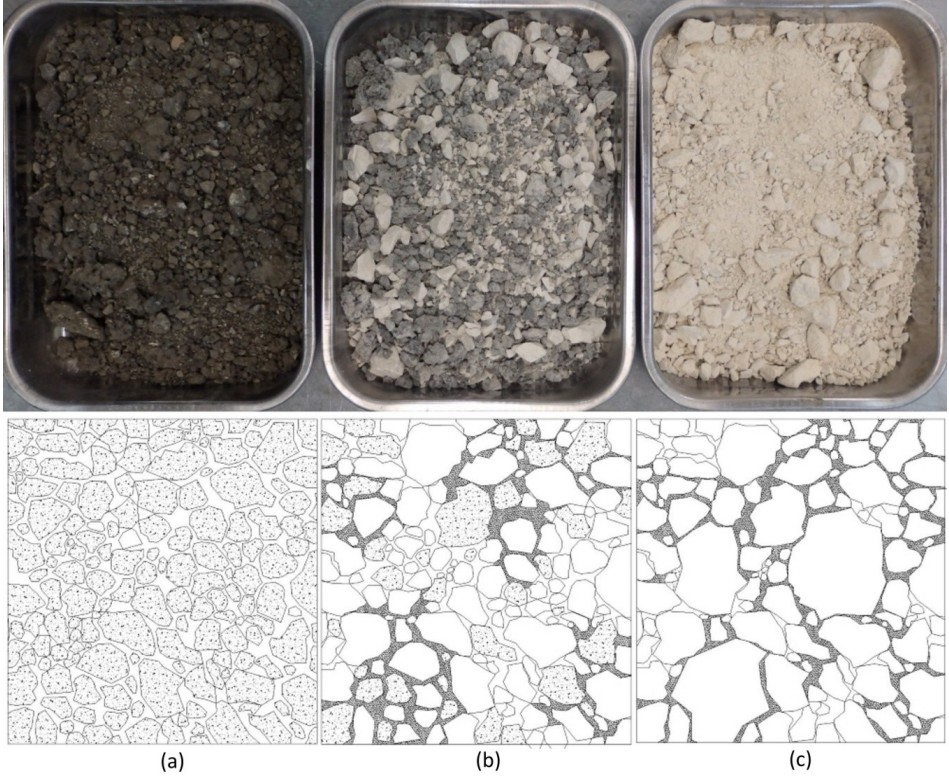

**Figure 2.** Investigated materials: (**a**) reclaimed asphalt pavement (RAP) (left), (**b**) 50/50 material (middle) (**c**) virgin aggregate (VA) (right).

The three investigated materials were appropriately sieved following the EN 933-1 and EN 933-2 standards to determine their gradation. Table 1 indicates that all three investigated materials were well graded gravels (GW) in accordance with the Unified soil classification system (USCS) and were classified as A-1-a in agreement with the AASHTO classification system. Figure 3 shows the determined material.

**Table 1.** Material classification.

| Property | RAP | 50/50 Material | VA |
|---|---|---|---|
| USCS | GW | GW | GW |
| AASHTO classification | A-1-a | A-1-a | A-1-a |
| Asphalt content | 4.2 | 2.1 | NA |

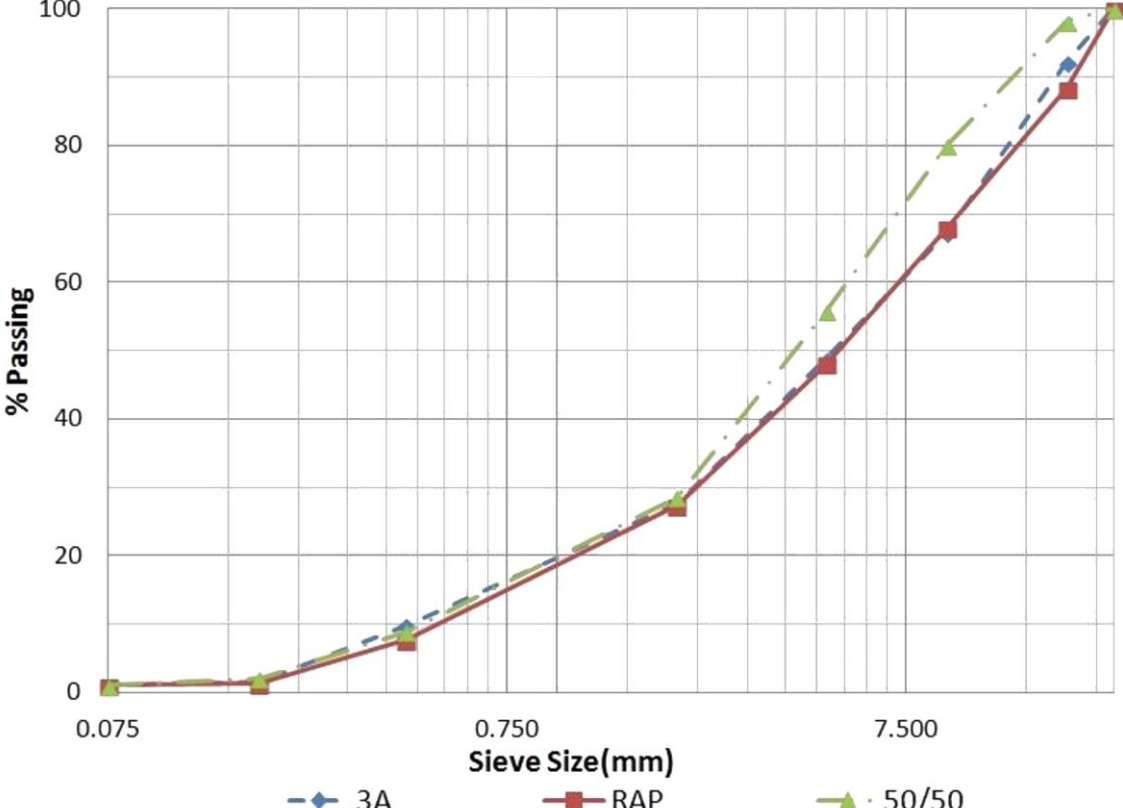

**Figure 3.** Material gradations.

As shown in Figure 3 the gradation curves for the three investigated material are very similar for aggregate sizes below 2 mm with an overall distribution of the materials similar overall. In order to achieve the close gradation curves, the mixtures were partially graded with emphasis on the fine materials as they are known to strongly influence laboratory modulus testing results.

### 4.2. Testing

Maximum Dry Weight (MDD) and Optimum Moisture Content (OMC) were determined utilizing the Modified Proctor test standard EN 13286-2-Type B mold. The dimensions of the compacted samples in accordance with the modified procedures were 120 and 150 mm for the height/diameter respectively. A 4.5 kg hammer was lifted and dropped from a height of 457 mm to impact the examined materials. The materials were compacted in five individual layers with 56 impacts per layer. Based on the results of the testing the OMC for the RAP material was 4.1% and the MDD equal to 2.079 kg/m$^3$. For the

50/50 material, the OMC was 5.9% and MDD equal to 2.225 kg/m$^3$. For the VA material, the OMC was 5.5% and the MDD was equal to 2.245 kg/m$^3$.

The T307 AASHTO Standard Method of Test for Determining the Resilient Modulus of Soils and Aggregate Materials was implemented to define the M$_r$ values of the examined materials. For the research the T307 AASHTO protocol for Material Type 1 was utilized, as it is valid for untreated granular base pavement materials, with 70% or less passing the 2 mm sieve and less than 20% passing the 75 μm sieve.

The investigated materials to be tested were dried and the appropriate amount of water was added to achieve the OMC. The sample material was then allowed to cure for 16 to 24 h in a sealed container and afterwards they were available for sample preparation. The M$_r$ test samples were prepared with a vibratory hammer with the each of samples compacted in a rubber lined split mold in six lifts with the prepared samples encased in porous stones (top/bottom) for testing. Final sample dimensions for M$_r$ testing were 150 mm (diameter) and 300–305 mm (height). The material for modulus testing was compacted to 95% to 96% of the modified proctor compaction results to meet specifications for base layer materials.

The T307 testing protocol requires an initial preconditioning of the to be tested sample, in order to ensure that the top/bottom surface are level and properly seated. A 103.4 kPa confining pressure was set and 1000 preconditioning loads with a 103.4 kPa maximum axial stress was applied. In continuation, the 15 stage load sequence per the AASTHO T307 was commenced. The sequence has defined: confining pressures—σ$_3$ (kPa), axial stresses—σ$_{max}$ (kPa), cyclic stresses—σ$_{cyclic}$ (kPa) and contact stresses—0.1σ$_{max}$ (kPa). In addition, the load sequence is a haversine load with a 0.1 s load duration that is followed in succession by a 0.9 s resting period (Figure 4). Each load sequence was applied 100 times with the final five load cycles recorded for further usage. Top mounted LVDT's were utilized to determine sample axial deformation. The testing sequence was carried out in a testing chamber capable of maintaining both the confining pressure and the ambient temperature. All resilient modulus tests were conducted at 25 °C (Figure 5).

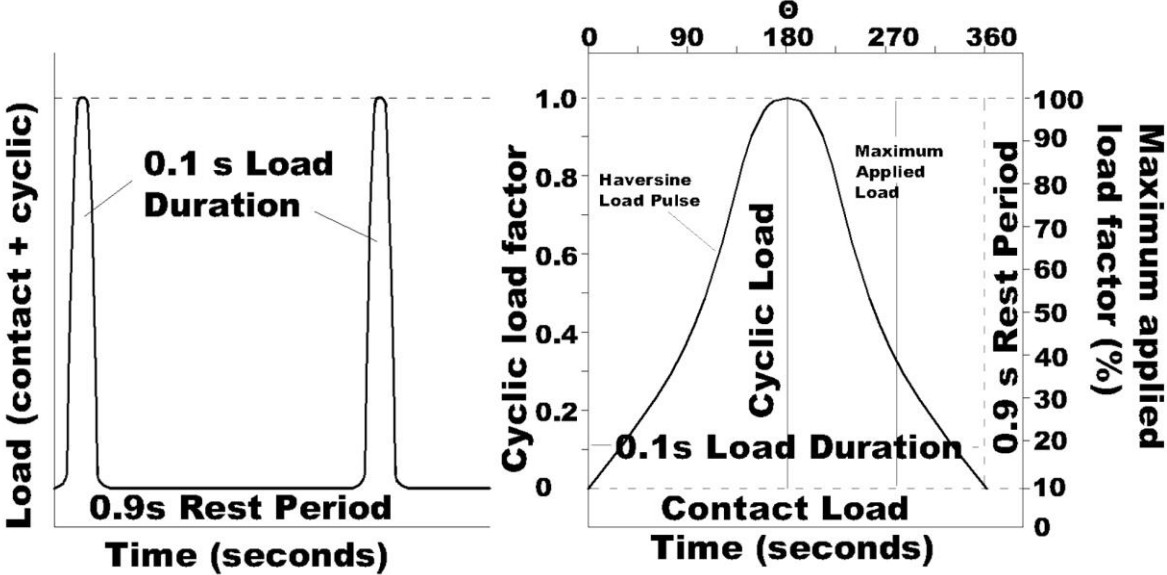

**Figure 4.** Resilient modulus (M$_r$) T307 loading protocol.

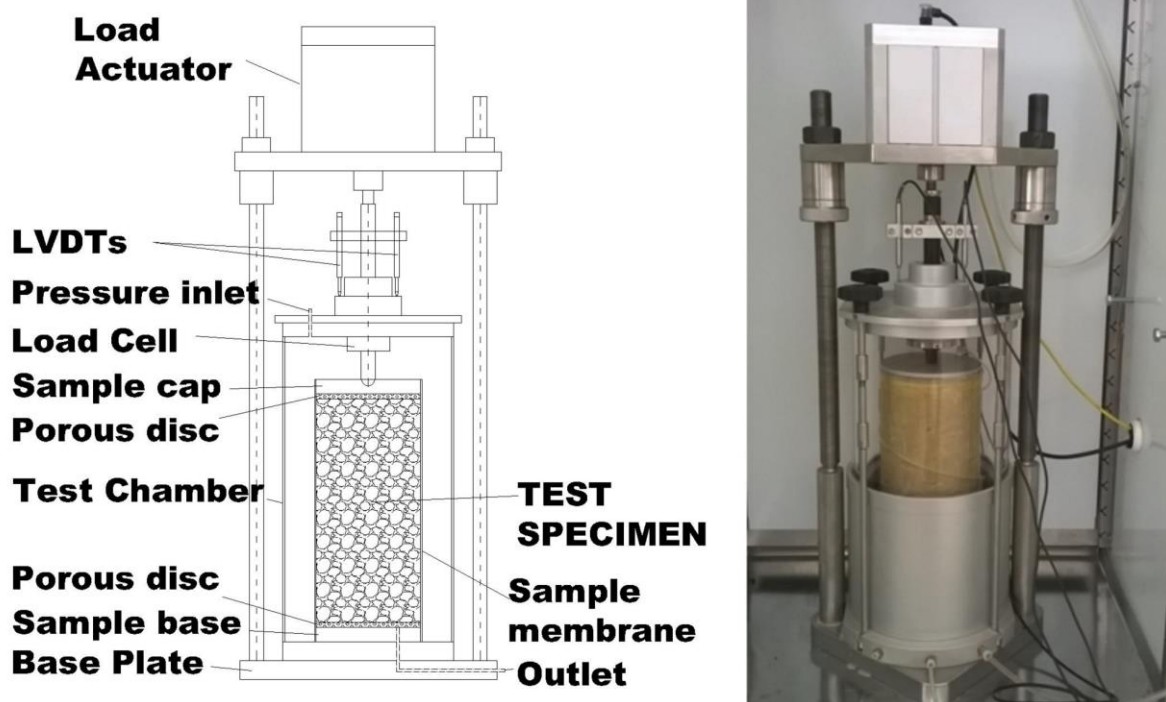

**Figure 5.** Resilient modulus (M$_r$) laboratory testing setup.

Presented in Figure 6 is an overview of the results from the laboratory tri-axial testing for the three investigated materials. This is the first phase of the current investigation, in which the tested RAP materials are compared against a VA suitable for base layer construction within flexible pavement structures. The results indicated that the materials containing RAP had comparable modulus values to the VA materials. The RAP material at confining stresses of 103.4 and 137.9 kPa produced modulus results approximately 10% to 15% lower than the examined VA at OMC. Similar results can be seen for the 50/50 material with the modulus results at confining stresses of 103.4 and 137.9 KPa producing results approximately 5% to 10% lower than the examined VA at OMC (Figure 7). The results from both the RAP and 50/50 materials exhibited overall lower coefficient of variation (COV) numbers when compared to the VA material, indicating a more stable material across the samples tested within the laboratory. This reduced variation was more evident at lower bulk stress levels as shown in the comparisons in Figure 6. Even though at the lower bulk stress levels the materials containing RAP exhibited slightly lower M$_r$ values when compared to the VA, they had less variation especially for bulk stress levels up to 300 KPa.

In Figure 7, the five confining stresses (20.7, 34.5, 68.9, 103.4 and 137.9 KPa) and curves for the deviator stress are shown. When investigating these curves for each of the three materials there are notable differences between the materials containing RAP in comparison with the VA materials. The slopes of the confining stresses for each of the two materials containing RAP are significantly lower in comparison to the VA material. The slope of the 100% RAP material shows that the material is, in general, less influenced by the deviator stress than a VA material. More specifically, at the minimum confining stress of 20.7 KPa, the effect of the deviator stress is only 12.9% of the comparable VA material variance. While as the confining stresses are increased to the maximum of 137.9 KPa the influence of the deviator stress is only 52.8% of the corresponding VA material variance. Correspondingly, the numbers for the 50/50 material are 27.6% and 82.9% when compared against the VA. Thus, it can be seen that as the RAP percentage increases in the mixture the material is less affected by the application of the deviator stresses and is influenced more significantly by the variations of the confining stresses. The importance of these differences is to be investigated in the second stage of the current investigation

regarding the applicability of current nonlinear constitute models to model the behavior of mixtures containing RAP.

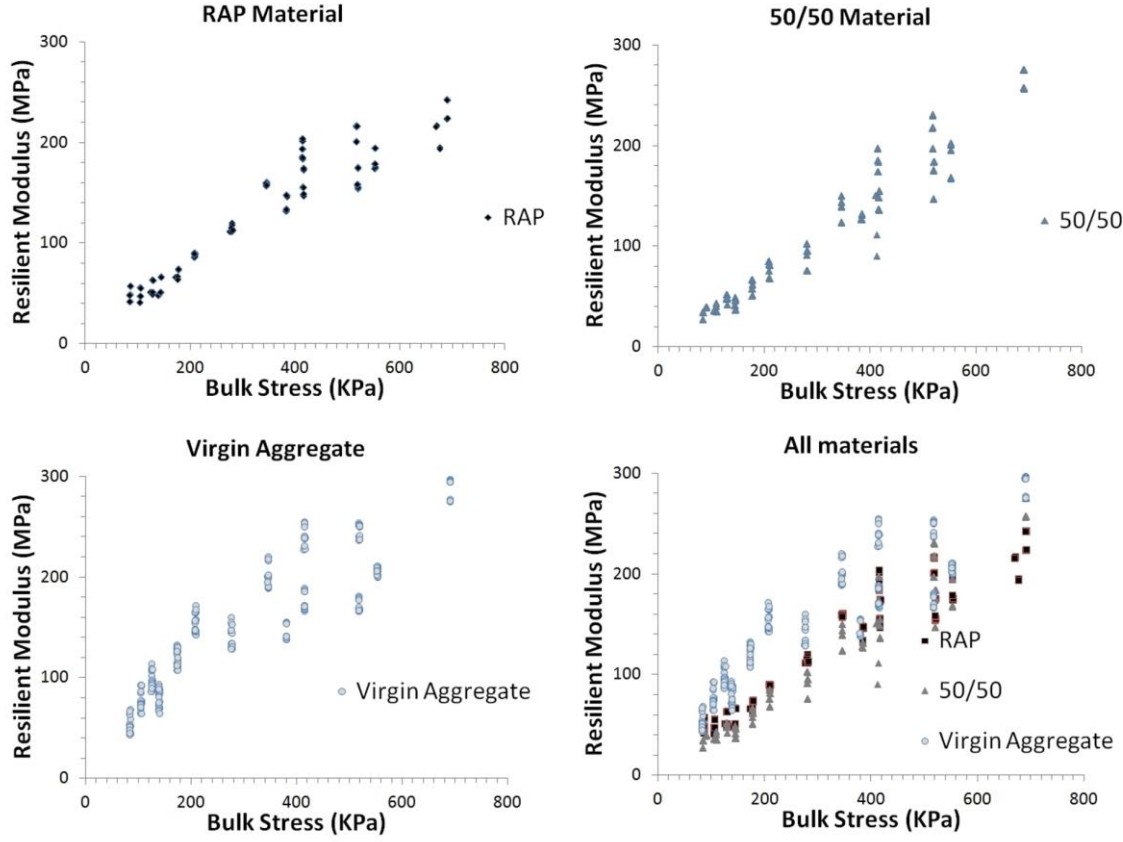

**Figure 6.** Resilient modulus ($M_r$) laboratory testing results—bulk stress.

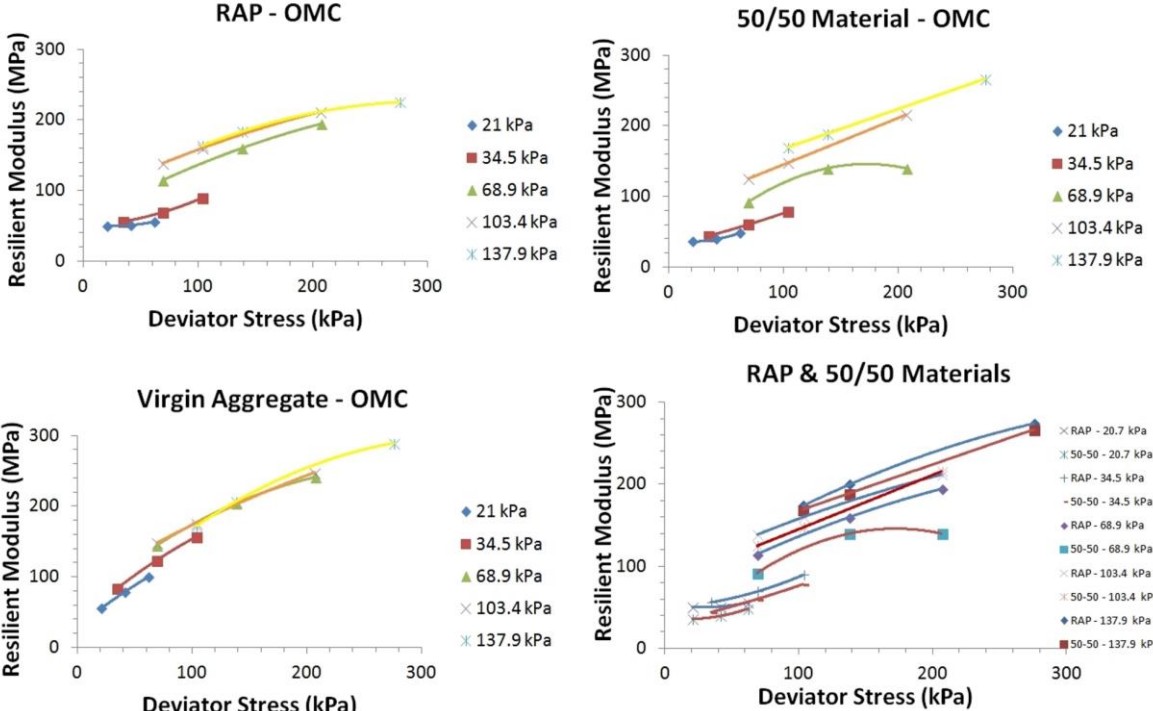

**Figure 7.** Resilient modulus ($M_r$) laboratory testing results—deviator stress.

## 5. Data Analysis

### 5.1. Modelling

The exported results from the laboratory determination of the resilient modulus were then utilized in a second stage of the study for a regression analysis in order to determine the variables for the four investigated constitutive models. Regression analysis was applied to determine the coefficient $k_1$, $k_2$ and $k_3$ variables (Table 2) for the K-θ, Uzan, Pezo and MEPDG models. An advanced nonlinear analysis of each of the investigated models was performed with a a 95% confidence level.

**Table 2.** Calculated regression constants.

| Material | Model | $k_1$ | $k_2$ | $k_3$ |
|---|---|---|---|---|
| RAP | K-θ | 516.96 | 0.786 | N/A |
| | Uzan | 785.93 | 0.568 | 0.234 |
| | Pezo | 1456.29 | 0.419 | 0.373 |
| | MEPDG | 505.95 | 0.625 | 0.484 |
| 50/50 | K-θ | 347.28 | 1.039 | N/A |
| | Uzan | 543.77 | 0.807 | 0.260 |
| | Pezo | 1377.56 | 0.525 | 0.529 |
| | MEPDG | 337.98 | 0.840 | 0.599 |
| VA | K-θ | 787.02 | 0.634 | N/A |
| | Uzan | 1878.32 | 0.175 | 0.461 |
| | Pezo | 1684.32 | 0.530 | 0.126 |
| | MEPDG | 738.00 | 0.301 | 1.025 |

The $k_1$, $k_2$ and $k_3$ constants, the predicted, measured and residuals values were exported for further analysis. With the constants determined for each of the investigated constitutive models, a comparison was then made to determine the goodness of fit of each of the investigated models. The determination of goodness of fit was determined by both the R-squared values ($R^2$) and the Root Mean Square Percentage Error (RMSPE).

### 5.2. Regression Analysis Results

Figures 8–10 present the predicted vs. measured $M_r$ values along with an equality line for comparison purposes. Figure 11 presents the predicted $M_r$ values and the residuals. Table 2 shows the $R^2$ and RMSPE per constitutive model. As can be seen in the figures for each of the materials and investigated models, the R-squared values were above 0.90 in all but one case, providing an initial indication that the models may be able to appropriately model the behavior of RAP materials. As can been seen in Figure 11 the variations in the predicted vs. the residual values for the VA has the most variation. While the RAP material overall had the least variation of the three investigated materials. As seen in Figure 11, the RAP material had the lowest variance in the residuals overall, while the VA material had the largest range of residuals throughout the spectrum of the stress levels. The 50/50 material exhibited behavior similar to the RAP material at low stress and modulus levels, while as the stress levels increased its behavior was more similar to the VA materials.

As can be seen in Figure 12, the K-θ model produced the lowest $R^2$ values for each of the three investigated materials overall, in addition to having the largest RMSPE values for both the VA and the 50/50 material. Particularly, for the VA the $R^2$ and RMSPE values (0.82 and 19.40), respectively, were significantly different from the other investigated models.

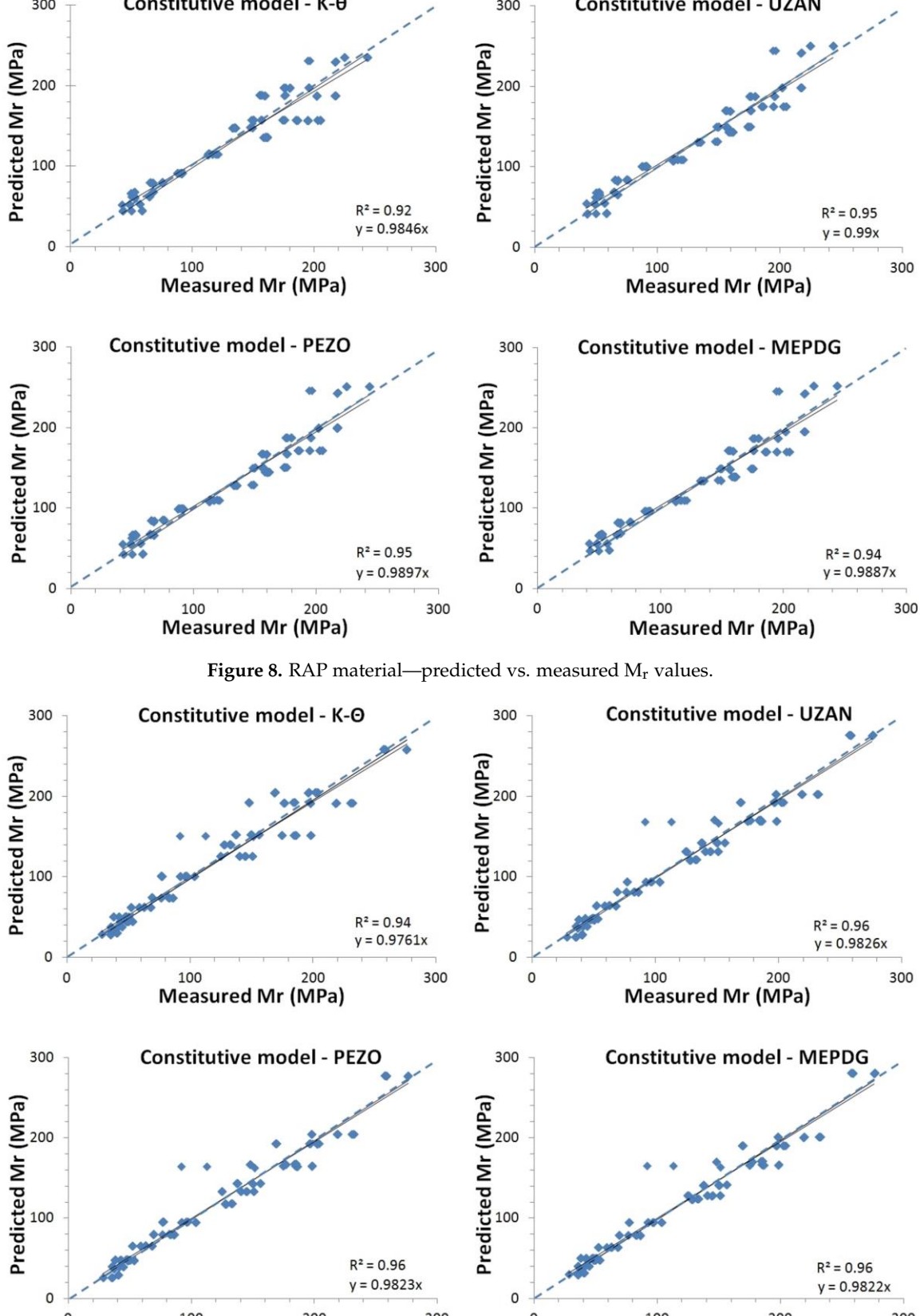

**Figure 8.** RAP material—predicted vs. measured $M_r$ values.

**Figure 9.** The 50/50 RAP-Aggregate material—predicted vs. measured $M_r$ values.

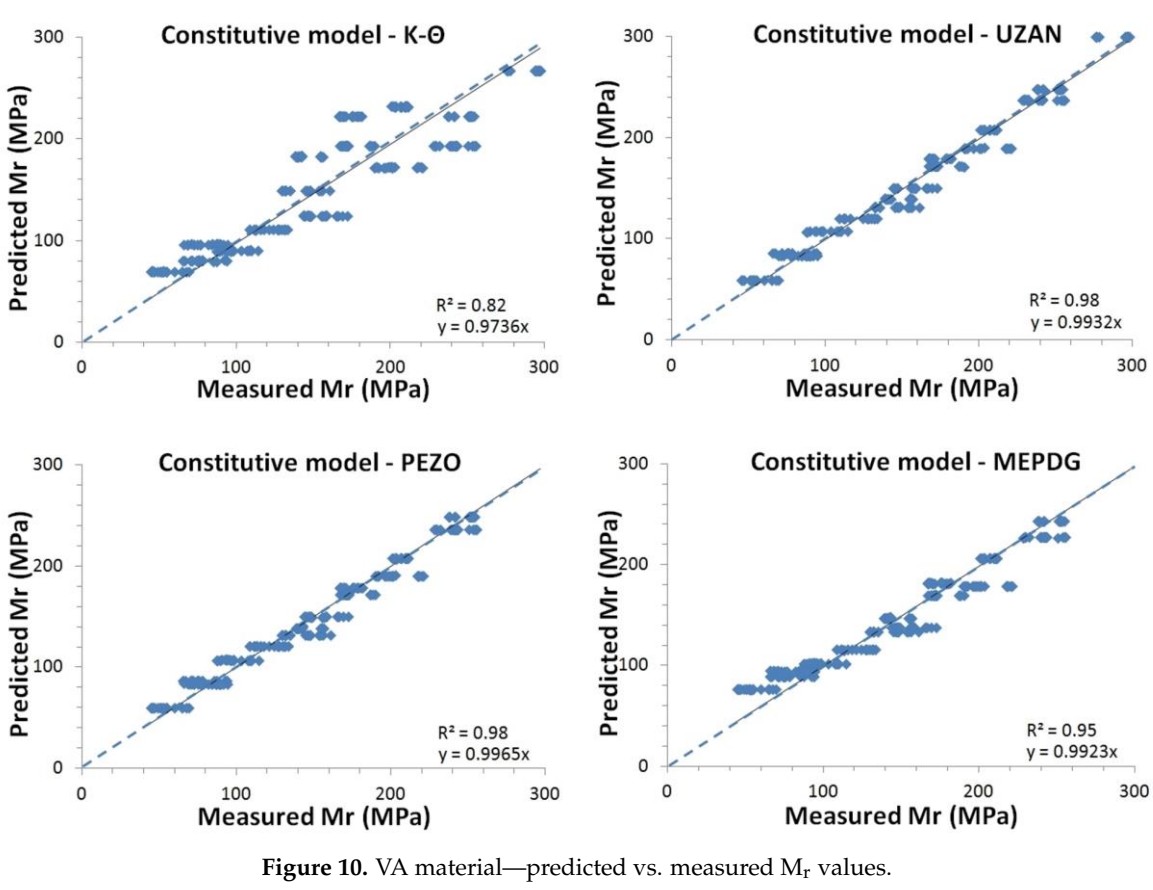

**Figure 10.** VA material—predicted vs. measured $M_r$ values.

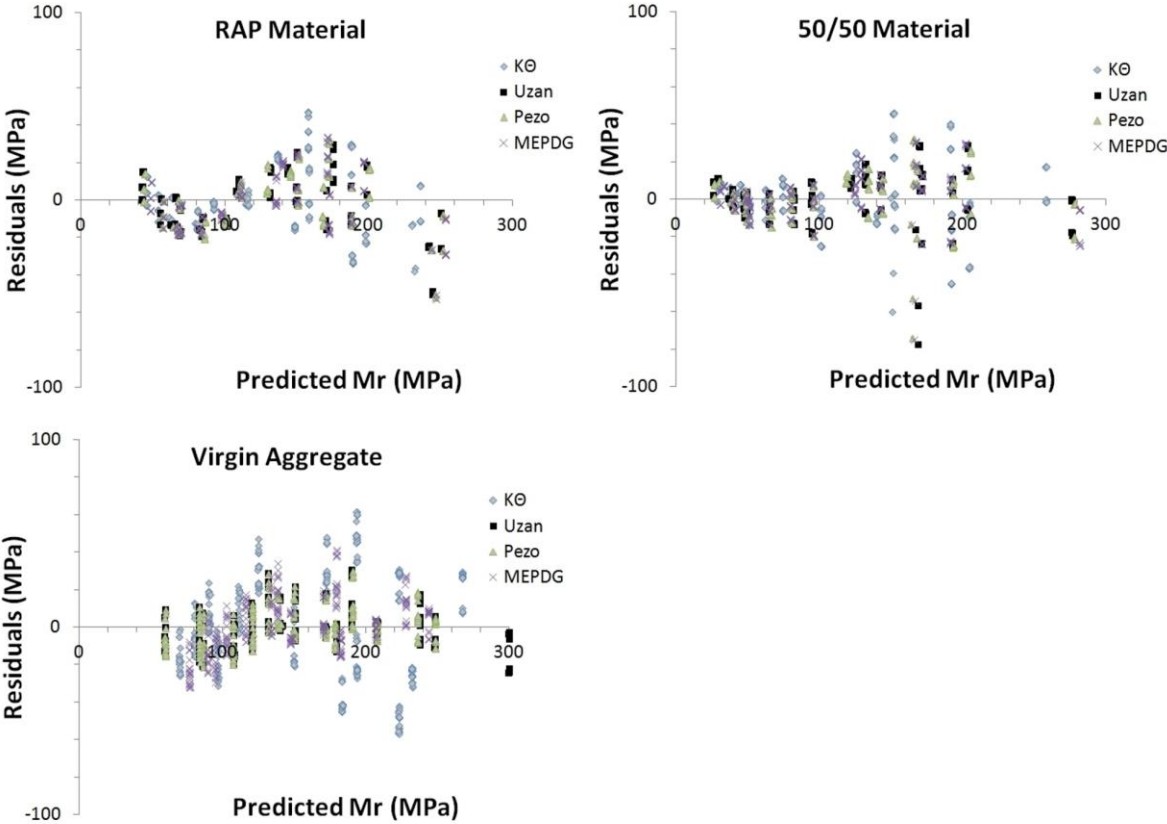

**Figure 11.** Predicted $M_r$ values and the residuals.

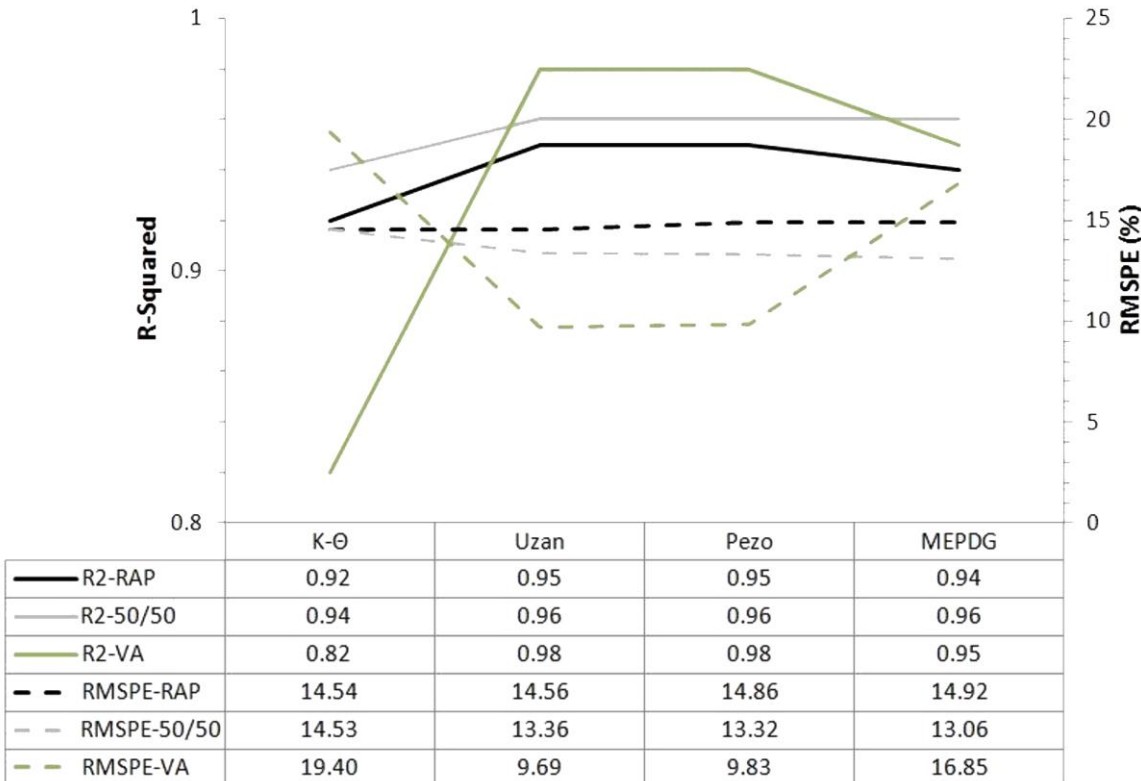

**Figure 12.** $R^2$ and the Root Mean Square Percentage Error (RMSPE)—constitutive models and investigated materials.

However, this difference was not noted in either the RAP or 50/50 material. Though the K-θ model produced slightly less accurate results, the variation was limited. The MEPDG model produced slightly more accurate results for the investigated 50/50 material, though the results were very close, in general, to the other models. The Pezo and Uzan models produced similar results for all three of the investigated materials. The investigated models are all based on the results of laboratory testing and incorporate many similar characteristics including the bulk stress, confining pressure, the deviator stress and the octahedral shear stress. Each of the models with exception the K-θ model utilized two of the mentioned characteristics. Based on this and the regression results, the variation in the $R^2$ and RMSPE between the models may have been reduced.

## 6. Discussion

The current research investigated three materials (1) a 100% RAP material extracted from a road section undergoing rehabilitation. The material was partially graded after being extracted from the road in order to meet gradation specifications (2) a 50% RAP/50% VA mixture and (3) a 100% VA material suitable for base layer construction. Initially, materials properties were determined, and tri-axial testing was conducted. The aim of the first section of the research was to investigate materials containing RAP material and comparing them against an accepted VA material for pavement base layer construction. The results indicated that the material containing RAP had resilient modulus values similar to the VA material.

With these results, the research proceeded to the second stage, which was to investigate the applicability of various nonlinear constitutive model's ability to characterize the resilient behavior of unbound aggregate materials containing RAP. More specifically the K-θ, Uzan, Pezo and MEPDG models were investigated for each of the three investigated materials. The regression constants for each of the models were determined and then a statistical analysis was performed to compare the

model predicted modulus values versus the laboratory determined resilient modulus results. The $R^2$ and RMSPE were both calculated for each of the four investigated models.

From the results of the analysis it was determined that all the models (the K-θ Pezo, Uzan and MEPDG model) produced similar results and based on the materials investigated, they could all potentially be utilized to model RAP material nonlinear behavior for the RAP materials investigated. For the investigated VA material, the K-θ model produced decreased accuracy in comparison to the other models, something that was not the case for the RAP materials investigated. This information provides evidence to support that RAP behavior can be modeled by existing nonlinear constitutive models that were examined, but more investigation is required. The ability to properly model materials containing RAP materials and the overall modulus results in comparison with the VA material provides evidence that the RAP materials investigated can be utilized in pavement design processes. In other words, their material resilient modulus properties are comparative to VA materials and the materials can be appropriately modeled for pavement design purposes.

Beyond the laboratory $M_r$ testing and the indications that the investigated models are able to properly define the $M_r$ characteristics, other factors should be further investigated for a more in-depth knowledge of the subject. These include investigating the maximum levels of RAP that can be included into base layer materials taking into account other considerations, such as permanent deformation, moisture susceptibility, optimum gradation curves. This issue should be more defined in order to more fully incorporate all characteristics of the RAP material into pavement design processes.

## 7. Conclusions

Overall, both the 100% RAP material and the 50% RAP/50% VA material produced modulus results similar to the investigated VA, especially at increased confining pressures. The RAP materials in regard to the resilient modulus exhibited greatly reduced influence from the deviator stress and the behavior of the materials containing RAP was mainly influenced by the level of the confining stresses.

In addition, it was found that all of the investigated models (i.e., K-θ, Uzan, Pezo and MEPDG), could equivalently describe the behavior of the materials containing RAP for pavement design purposes. This finding provides support for RAP inclusion as an unbound base material into both new and rehabilitation pavement projects suggesting a sustainability perspective.

To increase the accuracy of the prediction results towards the optimization of pavement design, new models that take into consideration the distinctive features of RAP materials should be developed. The current constitutive models, while often achieving excellent adaptation, were developed for the analysis of natural unbound material and do not take into account unique features such as the aging of asphalt binder etc. As a consequence, future research is needed to investigate other aspects including development of a permanent deformation model for RAP material in order to assess the allowable RAP percentages that can be included into flexible pavement structures as a base material. The finding of the current study that all the examined models predict slightly more accurately the moduli for the 50/50 material could suggest an initial maximum of 50% RAP to be set. Beyond this, the goal of pavement sustainability must deal with the pavement performance and construction quality. RAP is a material that can be fully recycled, and research is important to meet these goals and increase the sustainability of vital resources. In the future, based on research, distinct specifications should be set so that these materials can be more fully incorporated into the base layers of pavement structures.

**Author Contributions:** Conceptualization, C.P. and B.C.; Formal analysis, B.C.; Project administration, C.P. and B.C.; Supervision, C.P.; Validation, B.C.; Writing—original draft, B.C.; Writing—review & editing, C.P. and B.C.

**Funding:** This research received no external funding.

**Conflicts of Interest:** The authors declare no conflict of interest.

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
