# Peer review of "A Sustainability Perspective for Unbound Reclaimed Asphalt Pavement (RAP) as a Pavement Base Material"

_sustainability, doi:10.3390/su11010078_

Round 1

Reviewer 1 Report

This review report gives a few recommendations to improve the manuscript.

In general, the paper is well-written.

The introduction is well-written.

The literature survey is sufficient and introduces the important background for this study.

The methods used in the paper are laboratory studies. The chose methods are suitable for this study. The  Constitutive Modeling of Unbound Materials are properly described.

The analysis of the results is based on a comparison of the outcomes of the different  Regression Analysis. The analysis is properly described. A short discussion on the reasons for the differences between the outcomes of the models should be added (i.e. did you expect this outcome of the constitutive model when changing the properties in the model?).

The discussion of the results is sufficient, provided that the above-mentioned short discussion will be included.

A section with recommendations for practice should be added.

A section with future research (if you are planning future testing and/or future research on this topic) should be added. What remains to be investigated? Will you use your methods to optimize the mix design?

The conclusions are supported by the material presented in the paper.

Reference to figures and tables is done correctly. The resolution of the figures seems to be too low in the PDF; this needs to be improved.

To conclude: the authors are presenting interesting numerical work.

Author Response

Authors’ response (AR) to the Reviewers’ comments

Reviewers' comments:

Reviewer #1:

This review report gives a few recommendations to improve the manuscript.

In general, the paper is well-written.

The introduction is well-written.

The literature survey is sufficient and introduces the important background for this study.

The methods used in the paper are laboratory studies. The chose methods are suitable for this study. The  Constitutive Modeling of Unbound Materials are properly described.

The analysis of the results is based on a comparison of the outcomes of the different  Regression Analysis. The analysis is properly described. A short discussion on the reasons for the differences between the outcomes of the models should be added (i.e. did you expect this outcome of the constitutive model when changing the properties in the model?).

AR: Ok! Please see lines 383-388.

The discussion of the results is sufficient, provided that the above-mentioned short discussion will be included.

A section with recommendations for practice should be added.

AR: Ok! Please see lines 442-446.

A section with future research (if you are planning future testing and/or future research on this topic) should be added. What remains to be investigated? Will you use your methods to optimize the mix design?

AR: Ok! Please see lines 416-422.

The conclusions are supported by the material presented in the paper.

Reference to figures and tables is done correctly. The resolution of the figures seems to be too low in the PDF; this needs to be improved.

AR: Ok! The images have been reformatted.

To conclude: the authors are presenting interesting numerical work.

Reviewer 2 Report

Interesting paper. The objective of the study was to prove the sustainability of using RAP in base of the pavements an the authors performing tests to compare the performance of RAP materials against VA materials, and they made such comparison based on resilient modulus value; then the authors conducted regression analyses using a few different models (such as the one provided by Uzan). Then the authors performed s statistical analysis to compare the modulus values v.s. resilient modulus values, and proved that RAP is sustainable. The other clearly explained the objective and their research addressed their set objective. I have a few comments that the authors can address.

I agree that based on the findings of authors, RAP results in sustainability, but there are other variables that can influence the sustainability such as the costs associated with hauling RAP. Can authors mention this in the text and give a brief explanation?

It is great that the authors have conducted such thorough literature review, but I highly recommend to reduce the literature review to 2 pages rather than 4.

L2-3: The RAP expansion should not be capitalized. This should be fixed throughout the manuscript.

L163: there should be a hyphen between  "(RCA)" and "and", i.e., "(RCA)-and"

L274: Did authors mean appropriate amount of water?

L253: The authors would better to explain the reason behind the least variation of RAPM material.

Author Response

Authors’ response (AR) to the Reviewers’ comments

Reviewers' comments:

Reviewer #2:

Interesting paper. The objective of the study was to prove the sustainability of using RAP in base of the pavements an the authors performing tests to compare the performance of RAP materials against VA materials, and they made such comparison based on resilient modulus value; then the authors conducted regression analyses using a few different models (such as the one provided by Uzan). Then the authors performed s statistical analysis to compare the modulus values v.s. resilient modulus values, and proved that RAP is sustainable. The other clearly explained the objective and their research addressed their set objective.

 I have a few comments that the authors can address.

I agree that based on the findings of authors, RAP results in sustainability, but there are other variables that can influence the sustainability such as the costs associated with hauling RAP. Can authors mention this in the text and give a brief explanation?

AR: Ok! Please see lines 46-51.

It is great that the authors have conducted such thorough literature review, but I highly recommend to reduce the literature review to 2 pages rather than 4.

AR: Ok! The literature review section was altered.

L2-3: The RAP expansion should not be capitalized. This should be fixed throughout the manuscript.

AR: Ok! It’s done.

L163: there should be a hyphen between  "(RCA)" and "and", i.e., "(RCA)-and"

AR: Ok! Please see lines 167.

L274: Did authors mean appropriate amount of water?

AR: Ok! Please see lines 278-279.

L253: The authors would better to explain the reason behind the least variation of RAPM material.

AR: Ok! Please see lines 260-263.